# Immune Evasion in Cancer Metastasis: An Unappreciated Role of Monocytes

**DOI:** 10.3390/cancers17101638

**Published:** 2025-05-12

**Authors:** Marina R. Patysheva, Anastasya A. Fedorenko, Anna A. Khozyainova, Evgeny V. Denisov, Tatiana S. Gerashchenko

**Affiliations:** 1Cancer Research Institute, Tomsk National Research Medical Center, Russian Academy of Sciences, Kooperativny Str. 5, Tomsk 634009, Russiad_evgeniy@oncology.tomsk.ru (E.V.D.); t_gerashchenko@oncology.tomsk.ru (T.S.G.); 2Research Institute of Molecular and Cellular Medicine, Peoples’ Friendship University of Russia (RUDN University), Miklukho-Maklaya Str. 6, Moscow 115093, Russia

**Keywords:** monocyte, immune evasion, metastasis, progression, monocytic myeloid-derived suppressor cell, pre-metastatic niche, tumor hybrid cell, epithelial-mesenchymal plasticity, immunotherapy

## Abstract

Monocytes are important components of the immune system and critical regulators of cancer progression. Whereas monocyte participation in cancer development and response to therapy has been described extensively, their impact on metastasis remains underdiscussed. This review summarizes data concerning the influence of monocytes on metastasis formation during their presence in the circulation, primary tumor, and pre-metastatic niche, and highlights insights into the clinical relevance of targeting monocytes in metastasis prevention.

## 1. Introduction

Metastasis is a multistep process of cancer progression that includes tumor invasion, intravasation, the circulation of tumor cells in lymph or peripheral blood, extravasation to the area of pre-metastatic niches, and, finally, colonization [1,2]. The immune system is a major player at all stages of metastasis development [3].

Monocytes are circulating immune cells, which are involved in oncogenesis as precursors to tumor-associated macrophages (TAMs) and dendritic cells [4,5]. Additionally, monocytes play a patrolling role while in the bloodstream [6]. As immature patrolling cells, monocytes can adapt to clearing cellular debris and aid in reducing inflammation [6]. To date, it is clear that monocytes produce a variety of factors that affect the cells surrounding them, including tumor, stromal, and immune cells. Yet, the exact nature of this influence is largely unexplored.

Some of the immune cell populations are known to inhibit cancer progression, whereas others can support it. The current understanding of the role of the immune system in cancer progression is that the balance between the pro- and antitumor activity of immune cells plays a crucial role in the development of recurrence and metastases [7,8,9]. Monocytes and the cells they differentiate into can take on both these contrasting roles within the immune response. This dual function of monocytes arises from their ability to change their form and behavior depending on the surrounding context and environment [4,5]. In some contexts, these cells promote tumor growth via immune evasion; in others, they actively identify and kill tumor cells. However, our current understanding of the factors that drive these different contexts remains extremely limited.

In this review, we highlight evidence indicating that the functional state of monocytes plays a critical role in the metastatic progression of cancer. Insights from studies pointing to the contributions of monocytes to tumor cell survival and spread suggest that monocytes are important in the development of metastases from the bloodstream. The existing knowledge and further mechanistic studies could help identify monocytes and their functions as a new avenue for anti-metastatic therapy.

## 2. Formation and Characteristics of Monocytes

Monocytes are white blood cells developing from myeloid progenitors in the bone marrow in response to circulating factors [4,5,10]. It has also been reported that a population of bona fide undifferentiated monocytes can reside in the spleen [11,12,13,14]. Splenic monocytes and bone marrow monocytes possibly exhibit different functional profiles [15,16,17,18]. However, whether and how monocytes that enter the blood from the bone marrow differ from those that originate in the spleen has not been well studied, and this question remains to be explored. The lifespan of monocytes in peripheral blood is 1–7 days [19]. While circulating in the blood, monocytes can patrol the endothelial wall and phagocytize, and maintain the integrity of the endothelium [13].

Like other immune cell populations found in the blood, the monocyte population is heterogeneous. There are three major subsets of monocytes that have been formally identified by the Nomenclature Committee of the International Union of Immunological Societies [20]. In humans, this classification distinguishes classical (CD14++16−), intermediate (CD14+16+), and non-classical (CD14lowCD16+) monocytes based on the expression of the lipopolysaccharide receptor CD14 and FcγRIII/CD16 [20,21]. However, since CD14 is expressed on all monocytes, and CD16 is found in both non-classical and intermediate populations, clearly defining the criteria for distinguishing these populations is challenging. Murine models are often used to investigate the role of monocytes in the development of metastasis. However, it should be kept in mind that there are some important differences between mice and humans where monocyte nomenclature and markers are concerned. For mouse blood monocytes, a subdivision into three subsets similar to humans is proposed, that is, classical, intermediate, and nonclassical [20]. The classical monocytes show high Ly6C expression (Ly6C++), the intermediate monocytes show a high level of Ly6C (Ly6C+), and the nonclassical monocytes express a low level of Ly6C (Ly6Clow) [20].

Various studies have suggested adding other markers to the conventional panel to improve the definition and homogeneity of monocyte subsets. Based on protein marker detection, monocytes can be divided into different populations depending on the expression of receptors involved in cell maturation, adhesion, migration, and invasion, including CD62L, CCR2, CCR5, CXCR1, CXCR3, CXCR4, or inflammation-related CD64 (Fcγ receptor I), CD163, CD80, CD86, and others [22,23,24]. The study of immune populations down to single cells is now possible thanks to recent methodological advancements such as single-cell sequencing or an application of mass cytometry by time of flight (CyTOF). Since the introduction of these methods, various research groups began to distinguish new populations of monocytes in health and disease [25,26,27,28,29,30]. These single-cell studies have revealed discrepancies in defining the distribution of CD14++16−, CD14+16+, and CD14lowCD16+ monocyte subsets. Presumably, once sufficient data have accumulated, monocyte nomenclature will be updated.

The main function of monocytes is to supply key components of the immune system, such as macrophages and dendritic cells. This is where they make a significant impact on shaping innate and adaptive immune responses. According to the classical theory of R. van Furth, it was previously believed that macrophages and dendritic cells, including those in the tumor microenvironment, originate from the bone marrow and are formed during the migration of their precursors—monocytes—to peripheral organs [31]. It is now understood that, except for intestinal macrophages, most resident macrophages in peripheral organs and tissues originate from the embryonic yolk sac [32,33]. Under healthy conditions, the macrophage pool is composed of resident cells and replenished by monocytes [34]. In pathological conditions, circulating monocytes become the main source for replenishing the population of tissue macrophages and immature or tolerogenic dendritic cells, that lead to suppressing the immune response, facilitating tumor progression and metastasis [35]. From the blood, monocytes migrate to sites of acute or chronic inflammation following a chemokine gradient [10,36,37,38]. The main chemokine associated with monocyte trafficking into tissues is CCL2, also known as monocyte chemotactic protein-1 (MCP-1), along with CCL5 and CCL7 (monocyte chemotactic protein-3, MCP-3) [10,36,37,38,39].

Tumors and metastatic sites are recognized as chronic inflammatory foci [40]. The last ones are considered non-healing wounds, where cells constantly release inflammatory mediators, such as TNF-α, IL-6, TGF-β, and IL-1β, prostaglandins, chemokines, and extracellular vesicles, into the bloodstream [41]. The combined effect of these factors may contribute to a pro-tumor skew in the monocyte population and their migration to the tumor site [42,43,44]. Data from mouse models of transplanted mammary and lung tumors show that tumor-associated macrophages (TAMs) in the primary tumor originate from bone marrow or spleen progenitor monocytes [11,45,46]. In a mouse lung cancer model, monocytes take 14 days to transition into functional perivascular TAMs [47]. However, the contribution of monocytes as a source for the TAM population varies between mouse models. Notably, in lung tumors, monocyte-derived macrophages are distinguished by the expression of monocyte markers, including CCR2 (an embryonic cell marker indicative of monocyte lineage), VCAN, S100A6, CD52, FN1, SELL, and TREM2 [48]. A murine pancreatic ductal adenocarcinoma (PDAC) model contained both circulating inflammatory monocytes and tissue-resident macrophages as sources of TAMs [49]. In another murine PDAC model, monocytes were found to differentiate into an intermediate TAM population which then produced two distinct, long-lived TAM lineages [50]. Monocyte-derived TAMs, for example, TREM2+ TAMs, infiltrate the tumor core, whereas tissue-resident-derived TAMs, such as FOLR2+ TAMs, are predominantly located within the perivascular niche of the tumor stroma [51]. Nevertheless, the tumor replenishes all or part of its TAM pool by programming circulating monocytes.

Monocytes in cancer patients have a number of distinct characteristics. These include absolute monocyte counts, shifts in population composition, and changes in the cells’ transcriptomic profile [52,53,54,55,56,57]. Recent single-cell sequencing studies have revealed 3 to 18 distinct clusters of circulating monocytes in cancer patients [56,57,58,59,60] (Table 1). These alterations are clinically significant, demonstrating association with the cancer stage, disease outcome, and response to therapy [43,52,53,54,55,56,57,59,60,61].

## 3. Pro-Metastatic Role of Monocytes in the Bloodstream

Circulating monocytes are essential immunoregulatory agents in cancer patients. Mobilized from the bone marrow under the tumor’s influence, they promote immune evasion in the blood, maintain circulating tumor cells, and migrate to malignant sites to exert their tissue effects.

Tumor-derived factors are key mediators in the crosstalk between monocytes and tumor cells (Figure 1). The peripheral blood of cancer patients contains a lot of transitory monocytes migrating to the tumor site or the pre-metastatic niche following chemokine stimuli like CCL2, CCL5, CCL7, and CCL12 [37,62,63,64,65]. This often leads to elevated peripheral blood monocyte counts and has been described in several cancer types [5,66,67]. In patients with pancreatic cancer, high CCL2 in tumors is associated with a low CD8 T-cell infiltration and significantly decreases survival rates. CCR2 blockade in mice depletes inflammatory monocytes and macrophages, preventing liver metastasis [67] (Table 2).

In addition to recruiting, tumors can program circulating monocytes either directly or through intermediary cells by releasing various vesicles. One type is tumor microparticles, subcellular vesicles with cytosolic content, which are released into extracellular spaces and enter the circulation [68]. It has been demonstrated that alveolar and interstitial macrophages in the lungs can uptake tumor microparticles and secrete CCL2. This signal recruits monocytes, which, in turn, predominantly differentiate into pro-tumor macrophages that promote tumor growth through IL-6 secretion and fibrin deposition [68]. Tumors can also influence monocyte polarization by releasing vesicles that contain microRNA. In vitro experiments have shown that vesicles containing miR-203 can promote the differentiation of monocytes into pro-tumor macrophages when taken up by monocytes. A study conducted in colorectal cancer patients found that high expression of serum exosomes containing miR-203 is associated with an increased likelihood of developing distant metastases. Additionally, mice injected with miR-203-transfected colon cancer cells developed significantly more liver metastases than control animals [69].

When recruited or transformed by a tumor, circulating monocytes acquire pro-tumor properties. They can attract T-regulatory cells by secreting Il-10 and TGF-b, suppress the activity of CD8 effector T-cells, and produce anti-inflammatory cytokines [4,5,70]. Monocytes derived from patients with metastatic breast cancer resemble reprogrammed immunosuppressive monocytes from patients with severe infections in their surface and functional phenotype as well as in the transcriptome [71]. CD14+CD204+ monocytes isolated from the pulmonary vein are associated with aggressive clinical behavior in surgically treated patients with non-small cell lung cancer [72]. The immunosuppressive form of classical monocytes termed monocytic myeloid-derived suppressor cells (Mo-MDSCs) is characterized by their ability to suppress cytotoxic T cell and NK cell activation and support tumor progression and metastasis [73]. Splenectomy in non-small cell lung cancer-bearing mice depletes the subset of Mo-MDSCs in the blood and can even reduce and delay the seeding of distant metastases [74]. Mo-MDSCs, but not the total CD14+ monocyte population, are significantly increased in the peripheral blood of breast cancer patients with locoregional recurrence or metastasis to lymph nodes and visceral organs [71].

Thus, circulating monocytes can be recruited from the bone marrow and reprogrammed by tumor signals, contributing to the development of pathological conditions and supporting metastasis.

## 4. Monocytes and Circulating Tumor Cells: Cooperation Towards Tumor Hybrid Cells

Tumor cells that leave the primary tumor and enter the bloodstream are thought to be the key players in metastasis [75]. However, most circulating tumor cells (CTCs) die in the bloodstream due to anoikis, interaction with immune cells, or exposure to hemodynamic forces [76,77]. To overcome the stress caused by the bloodstream environment, CTCs interact with various components of the circulatory system. Unlike neutrophils and platelets, whose interaction with CTCs has been well studied and discussed by us earlier [78], the cooperation of monocytes with CTCs remains poorly understood. Some studies have reported a positive correlation between the number of monocytes and CTCs in peripheral blood in breast cancer [79,80], particularly in metastatic triple-negative breast cancer [80]. Moreover, the level of CTCs together with the monocyte–lymphocyte ratio is linked to a shorter overall survival in patients with metastatic breast cancer [81].

Evidence suggests that as monocytes travel through the bloodstream, they can support metastasis by enhancing the mechanical properties of CTCs necessary for their survival [81] (Figure 1). Co-culturing the polarized monocytic cell line U937 with prostate cancer cell lines 22Rv-1, DU145, and C4-2, which resemble CTCs in their nanomechanical properties, increases the softness, deformation resistance, and adhesion of tumor cells [81]. To successfully adhere to vascular endothelium, tumor cells form hetero-aggregates with monocytes due to the activation of TNF-α in both cell types [82]. The central role in the formation of these hetero-aggregates and their endothelial adhesion belongs to the Nf-κB-dependent upregulation of ICAM-1. Notably, the treatment of TNF-α-activated MDA-MB-231 tumor cells with the proteasome inhibitor MG132, which blocks Nf-κB activity, significantly reduces cell aggregation, while anti-ICAM-1 monoclonal antibodies significantly decrease the number of hetero-aggregates attached to the endothelium [83].

Recent studies have shown that tumor hybrid cells (THCs) contribute significantly to cancer progression [83,84,85,86,87]. They are formed by the fusion of tumor cells and normal cells, particularly monocytes. THCs exhibit phenotypic characteristics similar to those of the parental cells, allowing for their efficient identification in clinical samples. Due to the fusion with monocytes, tumor cells acquire the ability to evade immune defense and demonstrate a high rate of proliferation and migration, as well as epithelial-mesenchymal plasticity necessary for adaptation in blood flow conditions [84,85]. Hybrids between tumor cells and monocytes have been found in blood samples from patients with colorectal, lung, and prostate cancers [83,84,85,86]. Across all histological types of lung cancer, the percentage of monocyte THCs in blood in combination with the patient’s sex can predict metastasis, whereas in non-small cell lung cancer, the presence of more than one THC is associated with significantly shorter overall and cancer-specific disease-free survival after curative resection [84,86].

The mechanisms regulating the formation of hybrids between tumor cells and monocytes are poorly understood. THC formation is believed to depend on CD36 expression in tumor cells, with anti-inflammatory polarized monocytes predominantly involved in the fusion process [84]. However, it remains unknown whether tumor cells can fuse directly in the bloodstream or this process is limited to the primary tumor.

## 5. Primary Tumor: The Contribution of Monocytes in Cancer Cell Intravasation

Invasion and following intravasation of tumor cells into adjacent tissues, is one of the hallmarks of cancer and the first step towards metastasis [88]. Another way for the monocytes to mediate metastasis is to promote the replenishment of the population of TAMs during the invasion and intravasation stages. Here, we do not focus on the role of TAMs in the development of metastases, as this is a subject for a separate review, which has already been covered in other publications [89,90,91,92]. Nevertheless, we would like to highlight several mechanisms through which monocytes can influence the intravasation of tumor cells.

A mouse mammary tumor polyomavirus middle T-antigen breast cancer metastasis model (MMTV-PyMT) has shown that a unidirectional transition from migrating monocytes to perivascular macrophages is required for tumor cell intravasation [47]. Motile, circulating TAMs are newly recruited monocytes attracted by CCR2 signaling, which then differentiate into sessile perivascular macrophages (Figure 1). This unidirectional process is regulated by CXCL12 and CXCR4. Cancer cells induce a TGF-β-dependent increase in CXCR4 levels in monocytes, and CXCL12 expressed by perivascular fibroblasts attracts these mobile TAMs to blood vessels, entrapping mobile cancer cells. Once in a blood vessel, migrating TAMs differentiate into perivascular macrophages, promoting vascular leakage and intravascularization [47]. Following in vitro incubation with conditioned primary breast tumor culture media, monocytes from breast cancer patients secrete higher amounts of immunosuppressive, metastasis-related, and angiogenic cytokines [92]. By releasing IL-1β, monocytes promote E-selectin expression on endothelial cells, enhancing the adhesion of tumor cells to the vascular endothelium [93].

Therefore, although not primarily responsible for the initial stages of tumor cell intravasation, monocytes do actively engage in this process, contributing to metastasis.

## 6. Role of Monocytes in the Pre-Metastatic Niche Formation and Metastasis

### 6.1. Pre-Metastatic Niche

The pre-metastatic niche (PMN) represents a primed microenvironment in distant organs that facilitates the colonization and growth of metastatic cancer cells. Monocytes and their tissue-differentiated form—macrophages—play a pivotal role in establishing the PMN including under the antitumor treatment [93,94,95,96,97,98,99] (Figure 1). The PMN is enriched in the pathologically activated form of monocytes possessing immunosuppressive activity, which are also known as Mo-MDSCs. Long before it becomes a metastatic site, the primary tumor and stromal cells, such as cancer-associated fibroblasts (CAFs), produce CCL2 that stimulates the recruitment of monocytes and Mo-MDSCs to the PMN and their differentiation towards pro-tumorigenic macrophages [38,100,101]. Circulating monocytes express CCR2, whereas tissue-resident monocytes express minimal levels of this receptor. The recruitment of circulating monocytes to the PMN is followed by a more active secretion of the proteolytic enzyme MMP-9 by macrophages, which facilitates the establishment of the PMN [94]. It is shown that exposure to antibodies targeting human CCL2 inhibits the recruitment of circulating monocytes [38].

In the PMN, monocytes act together with neutrophils, also recruited by tumor-derived CCL2 stimuli, and facilitate tumor cell colonization. Pro-tumor neutrophils can promote tumor cell invasion through the release of proteolytic enzymes and reactive oxygen species (ROS) [102]. However, there are contrary data about monocyte-neutrophil cooperation against tumor cells. In a mice model of breast cancer, it was demonstrated that cooperation of CCR2 monocytes with tumoricidal TMEM173+ neutrophils targets disseminated tumor cells in the lungs and prevents metastatic outgrowth [103].

Monocytes contribute to the breakdown of endothelial tight junctions and the remodeling of the extracellular matrix, thereby enhancing the ability of cancer cells to extravasate and induce inflammatory processes [94]. It has been demonstrated that monocyte recruitment to the PMN is essential for tumor cell survival in murine models. For instance, an impaired monocyte function led to a decrease in the survival of injected tumor cells in the lungs [104]. Similarly, in a mouse model of metastatic melanoma (B16F10 model), the accumulation of CXCR3+ monocytes/macrophages in the lungs was essential for melanoma engraftment and the development of metastatic disease [105]. CCL12, another chemokine expressed in the PMN, attracts Mo-MDSCs to the site before metastasis, prompting them to secrete IL-1β, which stimulates E-selectin expression and facilitates tumor cell attachment to the endothelium [93].

In the context of PMN development, both monocytic and neutrophil-derived MDSCs are essential in creating a suppressive immune microenvironment. The main mechanisms of action include the depletion of T-cell activation by secreting arginase-1 or consuming cystine and cysteine [106,107,108]. Moreover, MDSCs generate reactive oxygen species that can inhibit T, B, and NK cells [108,109,110], as well as produce enzymes regulating adenosine metabolism and others [111]. Established by Mo-MDSCs, the immunosuppressive microenvironment primarily facilitates the survival of CTCs. MDSCs also produce various cytokines such as IL-6, IL-10, TGF-b, and VEGF involved in an epithelial-mesenchymal transition (EMT), while MMP-2, MMP-9, and ANG2 facilitate extravasation of CTCs [95,112]. Il-10 and TGF-b promote the recruitment and activation of T-regs, which sustain an immunosuppressive microenvironment. In addition, only Mo-MDSCs but not neutrophil-derived cells in the PMN can secrete CCL7, which stimulates the JAK/STAT3 pathway and facilitates the activation of dormant micro-metastatic cells. In mouse models of colorectal cancer, CCL7 and MDSCs inhibitors support the dormant status of metastatic cells and reduce metastasis or recurrence after radical surgery [113].

Primary tumors can sensitize the future site of metastasis by secreting factors that promote the colonization of the PMN by monocytes. Mouse models of Lewis lung carcinoma and B16 melanoma have demonstrated that primary tumor secretes TNF-α, TGF-β, or VEGF-α, which stimulate endothelial cell secretion of monocyte chemoattractants S100A8 and S100A9 in the lungs [114]. S100A8 and S100A9, along with VEGF-α and TNF-α, contribute to increased vascular permeability, facilitating monocyte extravasation into the PMN [115,116,117,118]. Next, immune cells are attracted to the PMN, leading to an increased production of inflammatory agents.

### 6.2. Metastasis

To date, multiple molecular effectors of monocytes have been described that may support metastasis formation. Monocytic cells carrying the α4-integrin receptor can bind to vascular cell adhesion molecule-1 (VCAM-1), highly expressed on the surface of mouse and human breast cancer metastatic cells in the lungs. When α4-integrin on monocytes is engaged, VCAM-1 transmits anti-apoptotic signals to breast cancer cells through the PI3K/Akt pathway, promoting tumor cell survival. VCAM-1 depletion using shRNA effectively inhibits metastasis formation [119]. In a mouse model of pancreatic cancer, pro-inflammatory monocytes secreting high levels of granulin glycoprotein promote the production of periostin by cancer stem cells. This process contributes to the formation of a fibrotic microenvironment that supports metastatic tumor growth in the liver [120]. Mouse bone marrow-derived monocyte-like cells recruited by CXCL12 are the primary source of thrombospondin-1, which contributes to the development of metastasis by inducing cytotoxic T-cell exhaustion and impairing vascularization [121]. Stabilin-1 is a scavenger receptor that is typical of pro-tumor macrophages [122]. The tissue repair-promoting stabilin-1+Ly6Chi mouse monocyte subset has been shown to expand following the resection of the primary tumor and to promote lung metastasis originating from circulating tumor cells [123]. Immunophenotyping in mice demonstrated that bone metastasis-associated macrophages express high levels of CD204 and IL4R. Furthermore, monocyte/macrophage IL4R ablation significantly inhibits bone metastasis growth, and IL4R null mutant monocytes fail to promote bone metastasis outgrowth [124].

Monocytes identified in melanoma brain and leptomeningeal metastases have transcriptomic signatures distinct from monocyte clusters in the primary tumor [60]. In metastatic tissue, monocytes exhibit upregulation of numerous pro-tumorigenic and metastasis-promoting factors such as migratory chemokines (CXCL3, IL8, ILB, CXCL2, CXCL5, CCL20, and CCL3), tumor cell proliferation and survival molecules (AREG, EREG, and SPP1), and cancer progression-associated scavenger receptor CD163 [60,125]. Monocytes from metastases are also enriched in pathways associated with NOD receptor signaling, further supporting the notion of an enhanced monocyte migration from the blood to metastasis sites.

### 6.3. Contribution of Non-Classical Monocyte Populations

Most of the above data on the contribution of monocytes to the development of metastases concerned the classical population. Since this population accounts for 85–95% of the total monocyte pool, studies rarely focus on minor populations, such as the non-classical monocytes. However, unlike classical monocytes, the non-classical “patrolling” subset can inhibit cancer metastasis. In particular, IFN-γ induces intermediate monocytes, which inhibit melanoma metastasis by activating NK cells through FOXO1 and IL-27 in B16F10 mouse models [126]. Monocytes migrate to the metastatic niche and eliminate tumor cells either through direct engulfment or by activating cytotoxic NK cells [127]. Vesicles from poorly metastatic tumors induce a shift in macrophages toward a proinflammatory state marked by TRAIL expression. These TRAIL-positive macrophages compete with NK cells for tumor destruction. Importantly, non-classical CX3CR1high monocytes can act as “intravascular housekeepers” that scavenge tumor microparticles and vesicles, thereby protecting against metastatic colonization [128]. As shown in mouse models, non-classical monocytes scavenge tumor material and inhibit the attachment of tumor cells to the lung microvasculature. In addition, they promote the recruitment and activation of NK cells. Thus, transferring CX3CR1high monocytes into CX3CR1-deficient (Cx3cr1−/−) mice prevents tumor invasion and reduces tumor metastasis to the lungs [128].

Hence, classical monocytes and their immunosuppressive form—Mo-MDSCs—are the major mediators in establishing a favorable microenvironment in the future metastatic site and potentiating metastasis outgrowth. In contrast, non-classical monocyte subsets have the ability to eradicate metastases, which may be exploited in therapeutic applications.

## 7. Targeting Monocytes to Suppress Metastasis

Monocytes are essential for maintaining suitable conditions for metastatic dissemination across various levels. In tumors and the PMN, newly recruited myeloid cells modulate the local microenvironment by secreting chemokines, inflammatory factors, and matrix-degrading enzymes. This facilitates the migration of tumor cells through the vasculature and metastatic outgrowth. In circulation, monocytes maintain immunosuppressive properties to protect tumor cells from cytotoxic T and NK cells, enhance the mechanical resilience of CTCs, or form hybrid cells with them.

Four types of therapeutic strategies for targeting myeloid cells and preventing metastasis can be distinguished: blockade of monocyte recruiting, reprogramming of monocytes towards pro-inflammatory and anti-metastatic phenotypes, using monocytes as therapeutical carriers, and targeting Mo-MDSCs to prevent metastasis (Table 3).

### 7.1. Blockade of Monocyte Recruiting

During carcinogenesis, malignant cells release CCL2 to recruit CCR2-positive classical monocytes and Mo-MDSCs to promote tumor growth and progression. The blockade of monocyte-recruiting stimuli by targeting CCL2-CCR2 signaling has long been considered the most promising strategy for inhibiting the metastatic process [129,130]. Although many studies have explored clinical applications of CCL2 or CCR2 antagonists, they have not produced the expected favorable outcome. The studies demonstrated that targeting the CCL2-CCR2 pathway provides no considerable immediate or long-term benefits in either early-stage or metastatic tumors, suggesting the presence of compensatory mechanisms [129,130].

### 7.2. Monocyte Reprogramming

Undoubtedly, it is crucial to reevaluate therapeutic strategies targeting monocytes. One potential approach is to focus on methods for reprogramming these cells. Monocytes are well known for their ability to exhibit both pro-inflammatory and antitumoral activity and function depending on the context and environment. As shown in triple-negative patient-derived xenograft (PDX) models of breast cancer, pre-metastatic CCL2 niches are enriched in IFNγ-producing CCR2+ monocytes only in nonmetastatic areas. This leads to the upregulation of Tmem173/STING in neutrophils and enhances the targeting of disseminated tumor cells in the lungs by neutrophils [103]. Interestingly, neutrophils become TMEM173-positive only if the lung microenvironment is infiltrated by IFNγ-producing CCR2+ monocytes, indicating that the monocyte immune profile can control metastatic progression [103].

Thus, the main goal of reprogramming is to change the phenotype of tumor-recruiting monocytes towards anti-tumoral activity. A potential solution could involve using extracellular vesicles, exosomes, or transcriptional regulators to activate the desired monocyte phenotype. For example, tumor-derived extracellular vesicles from human squamous head and neck cancer and lung adenocarcinoma cell lines can interact with primary monocytes and induce their activated phenotype, which is characterized by the secretion of pro-inflammatory IL-1β and TNF-α [143].

MiRNAs are also known to regulate the functional activity of monocytes and macrophages. MiR-155, the most promising microRNA for reprogramming monocytes and macrophages, enhances the production of proinflammatory cytokines and induces the repolarization of macrophages from anti-inflammatory to proinflammatory [133]. MiR-155 knockdown in the myeloid compartment of transgenic mice accelerates tumor growth and significantly reduces the proportion of CD11c+ inflammatory monocytes [134]. Another potential therapeutic agent is miR-126. There is evidence that the level of miR-126 is downregulated in metastatic tissues compared to primary tumors in breast cancer patients. Moreover, it was demonstrated that duplex miR-126/miR-126* suppresses the recruitment of inflammatory monocytes in an SDF-1α-dependent manner by inhibiting the expression of CCL2 in the tumor stroma [135]. Treatment of xenograft mice with miR-126/miR-126* inhibits lung metastasis by breast tumor cells [144].

Another approach to monocyte programming is to shift the phenotype of monocytes to the non-classical state. To this end, exosomes derived from non-aggressive melanoma cells with low metastatic potential may serve as an immunotherapeutic solution, as they demonstrate anti-tumor properties. The exosomes are internalized by bone marrow monocytes and promote their differentiation into non-classical monocytes. Non-classical monocytes can migrate to the metastatic niche and eliminate tumor cells through direct engulfment or by activating cytotoxic NK cells [127].

Evidence shows that treating classical monocytes with small-molecule activators of NOD2 induces CCR2 expression and promotes the differentiation of classical monocytes into an inducible non-classical phenotype with non-canonical CX3CR1lowCCR2+MHCIIhigh markers. In mice, inducible non-classical monocytes can migrate into both vascular and extravascular tumor microenvironments via the CCR2/CCL2 axis, where they release CCL6 to recruit NK cells that promote tumor lysis independent of T and B lymphocytes, attenuating metastasis [136].

### 7.3. Monocytes as Therapeutical Carriers

The ability of monocytes to abundantly infiltrate tumor tissues and differentiate into macrophages is used in the development of monocyte-based delivery systems. Due to their phagocytic activity, monocytes can uptake and deliver anticancer agents, including nanoparticles, chemotherapeutics, and gene constructs [145]. Monocyte-based delivery systems have several advantages, such as enhanced tumor targeting, cargo protection during delivery, and biocompatibility. The main principle of this therapeutic approach involves modifying inflammatory monocytes with drug-loaded nanoparticles and reintroducing them into circulation, where they are recruited to the PMN site. The drug is released as monocytes differentiate into macrophages, due to the drug being linked to legumain, which is minimally expressed in monocytes but is significantly upregulated in macrophages. Cytotoxic mertansine, conjugated to poly(anhydride) with a legumain-sensitive peptide and legumain protease, can inhibit lung metastasis of breast cancer in mice. This effect is achieved by assembling the drug self-assembled into nanoparticles and loading them into monocytes (M-SMNs) [137].

There is another, novel immunotherapeutic approach to preventing metastasis called CAR-monocyte/macrophage (CAR-M) therapy. CAR-M therapy has evolved from CAR-T technology, offering enhanced precision due to the active infiltration of myeloid cells, particularly monocytes and their macrophage descendants, into solid tumors. This contrasts with T cells, which often fail to infiltrate the tumor microenvironment. This personalized therapy begins with isolating primary monocytes from the patient’s blood, which are then modified with a chimeric receptor that is specific to the desired antigen and relies on unique viral or non-viral methods for transduction. The resulting CAR-M cells are reintroduced to the patient’s blood by re-infusion. At the moment, CAR-M therapy for up to 12 targets can be designed, and this approach has already been tested against HER2 in vivo. Administration of CAR-M cells in a mouse model of ovarian cancer resulted in tumor control and suppression of metastases [138]. CAR-M therapy was also tested in HER2-overexpressing mouse xenograft models of colorectal and breast cancer and demonstrated synergy with anti-PD1 therapy [139].

### 7.4. Targeting Mo-MDSCs

While exploring the contribution of monocytes in cancer dissemination and the current innovations in treatment strategies, it is important to consider the Mo-MDSCs immunosuppressive population. These cells are essential for the establishment of an immunosuppressive microenvironment in the PMN. Potentially, therapy can be focused on targeting Mo-MDSCs, especially inhibiting their activity and recruitment into the PMN. Mo-MDSCs-targeting approaches are multifaceted and have been extensively discussed [140,141,142]. For instance, several Mo-MDSCs inhibitors, for example, X-682, a selective CXCR1/CXCR2 inhibitor, have been shown to suppress metastasis [140].

Since monocytes play a key role in cancer initiation and progression, they have become a focus of antitumor therapy. Monocytes are abundant in the blood and have a strong ability to infiltrate tissues, which makes them the primary target for therapeutic strategies aimed at using monocytes as antitumor agents rather than disabling them. Non-classical monocytes in particular can exhibit anti-metastatic activity. This ability can be leveraged as a therapeutic strategy if classical monocytes are reprogrammed into non-classical ones for tumor cell elimination. In contrast, the preferred pharmacological approach for Mo-MDSCs involves depleting their population or inhibiting their function. Moreover, since monocytes are continuously renewed and lack long-term immunological memory, it is important to highlight that this type of therapy will likely require extended treatment. Nevertheless, all modified monocytes will be eliminated quickly, while the physiological population is fully replenished, minimizing side effects.

## 8. Concluding Remarks

Metastatic dissemination is a complex multistep process promoted by multiple factors. The main cause of cancer progression is the accumulation of somatic mutations and epigenetic changes, improving the ability of tumor cells to migrate, differentiate, and survive. However, the development of metastases largely depends on the availability of an optimal environment in the bloodstream and the secondary sites. Monocytes are key players in providing the necessary conditions that support the metastatic process both in the bloodstream and locally. The most critical effect of monocytes on metastasis takes place during the PMN formation stage when monocytes and their immunosuppressive subtype—Mo-MDSCs—remodel the extracellular matrix and facilitate extravasation, attachment, and survival of CTCs. At the same time, non-classical monocytes demonstrate the opposite function. They can prevent cancer metastasis by scavenging tumor cells, inhibiting their attachment to the microvasculature, and recruiting cytotoxic lymphocytes. A critical question in advancing monocyte-focused treatments for metastasis prevention is how to suppress pro-tumor populations while strengthening those that fight metastasis. Resolving this issue would be a meaningful step toward utilizing pro-tumor monocytes to help cancer patients.

## Figures and Tables

**Figure 1 cancers-17-01638-f001:**
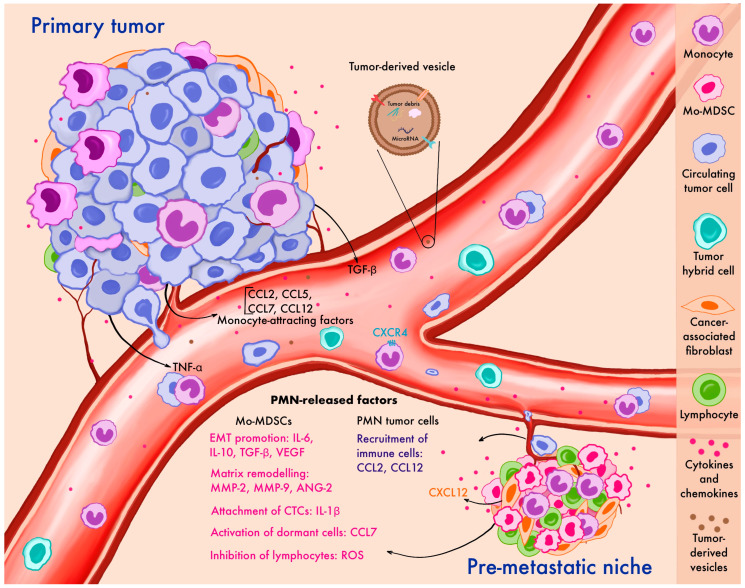
Contribution of monocytes to metastatic dissemination at various stages of the metastatic cascade: the primary tumor, circulation, and pre-metastatic niche formation. Primary tumor cells release factors that recruit monocytes, support their survival in circulation, and promote their migration to the pre-metastatic niche, where monocytes facilitate metastatic outgrowth. CTCs—circulating tumor cells, EMT—epithelial-mesenchymal transition, Mo-MDSCs—monocyte myeloid-derived suppressor cells, PMN—pre-metastatic niche, ROS—reactive oxygen species, TNF-α—tumor necrosis factor alpha, TGF-β—transforming growth factor beta.

**Table 1 cancers-17-01638-t001:** Monocyte populations in cancer patients identified by single-cell RNA sequencing.

Cancer Type	Monocyte Clusters	Markers	Refs
Lung cancer (N = 7)	Mono_1_ Mono_2_ Mono_3_	*1. CD14*, *FCN1*, *OLR1* *2. CDKN1C*, *LILRB2*, *ITGAL* *3. S100A8*, *S100A9*, *CSF3R*, *PROK2*, *VCAN*	[55]
Colorectal cancer (N = 5)	Classical Mono Intermediate Mono FCGR3A+ Mono TR cell activation related Mono Granulocyte chemotaxis Undefined Mono Granulocyte-like Mono S100A12high granulocyte-like Mono INFα-activated Mono	Not specified	[54]
Breast cancer Lung cancer (N = 10)	FCGR3A Mono IL1B Mono CXCL10 Mono CLEC10A Mono FOS− Mono FOS+ Mono	*FCGR3A* *IL1B* *CXCL10* *CLEC10A* *FOS−* *FOS*	[56]
Ovarian cancer (N = 3)	CD16 Mono S100A^high^ Mono MHC^high^ Mono	*FCGR3A* *S100A8, S100A9, S100A12* *HLA-DR, HLA-DQ, HLA-DAQ*	[57]
Melanoma (N = 66)	1.Subgroup 0 2. Subgroup 1 3. Subgroup 2 4. Subgroup 3 5. Subgroup 4 6. Subgroup 5 7. Subgroup 6 8. Subgroup 7 9. Subgroup 8 10. Subgroup 9 11. Subgroup 10 12. Subgroup 11 13. Subgroup 12 14. Subgroup 13 15. Subgroup 14 16. Subgroup 15 17/18. Subgroup 16/17	1. *RPS26*, *RPL7*, *RPS3A*, *RPL21*, *EEF1A1* 2. *S100A12*, *S100A8*, *S100A9* 3. *IFI2*, *IFIT3*, *ILRN1*, *CXCL10* 4. *CXCL9*, *HLA-DRB5*, *HLA-DQA1* 5. *RPS26*, *RPL7*, *RPS3A*, *RPL21* 6. *CLEC5A*, *TGFB1*, *FN1*, *YWHAH*, *FBP1* 7. *RSAD2*, *CSTB*, *HSPH1*, *DNAJB1* 8. *FCGR3A*, *CDKN1C*, *NAP1L1*, *SPN* 9. *FOS*, *ZFP36*, *BTG1*, *GPR183*, *IER3* 10. *CD163*, *IFI6*, *RNASE1*, *C1QC*, *C1QA* 11. *AREG*, *EREG*, *THBS1*, *THBD*, *TIMP1* 12. *PLIN2*, *IL8*, *HSP90AA1* 13. *TRAC*, *TRBC2*, *GZMA*, *CCL5* 14. *NFKBIA*, *EGR1*, *NFKBIZ*, *IL1B* 15. *FABP5*, *APOC1*, *SPP1*, *HMOX*, *APOE* 16. *CXCL3*, *CXCL2*, *CXCL8* 17/18. *C1QB*, *IFI27*, *NUPR1*	[58]

**Table 2 cancers-17-01638-t002:** Interaction between tumor cells and monocytes underlying the metastasis process.

Location	Cancer Secreted Factors	Monocyte Cell Response	Cancer Types/Models
Bloodstream	CCL2, CCL5, CCL7, and CCL12	Recruitment of monocytes via CCR2	Lung, pancreatic cancer
	Tumor microparticles	Recruitment of monocytes via CCR2 Pro-tumor macrophage differentiation	Colorectal cancer
	TNF-α	Formation of hetero-aggregates between tumor cells and monocytes via upregulation of ICAM-1	Prostate and breast cancers
	CD36	THC formation with anti-inflammatory polarized monocytes	Colorectal, lung, and prostate cancers
Primary tumor	TGF-β, CXCL12	Recruitment of monocytes via CCR2, CXCR4	Breast cancer
	IL-1β	Monocytes promote tumor cell adhesion to endothelial via upregulation of E-selectin	Colorectal cancer
PMN	CCL2	Recruitment of monocytes and Mo-MDSCs and their differentiation towards pro-tumorigenic macrophages	Colorectal, breast cancer, melanoma
	TNF-α, TGF-β, VEGF-α	Monocyte extravasation into the PMN via S100A8/A9	Lung metastasis of LLC, B16 mice tumors
	CCL12	Recruitment of Mo-MDSCs and promoting them to secrete: 1. IL-1β, IL-6, IL-10, TGF-b, and VEGF involved in EMT; 2. arginase-1 and ROS which creates a suppressive immune microenvironment; 3. CCL7 activating dormant micro-metastatic cells	Metastatic melanoma, colorectal cancer
Metastatic site	VCAM-1	Monocytes upregulate migratory chemokines (CXCL3, IL8, ILB, CXCL2, CXCL5, CCL20, CCL3), tumor cell proliferation factors (AREG, EREG, SPP1) throught α4-integrin receptor binding	Breast cancer metastatic cells, pancreatic cancer, melanoma brain and leptomeningeal metastases

Note: LLC—Lewis lung carcinoma; B16—melanoma.

**Table 3 cancers-17-01638-t003:** Strategies for targeting monocytes to suppress metastasis.

Strategy	Approach	Mechanism	Effect	Current Stage	Refs
Blockade of monocyte recruitment	Monoclonal antibodies or antagonists	Target CCR2 or CCL2	Metastasis inhibition in mice models of hepatocellular carcinoma	Pre-clinical	[129,130]
			CCR2^+^ monocytes decreasing	Phase II clinical trials	[131]
Monocyte reprogramming strategies	Tumor-derived extracellular vesicles (TEVs)	Releasing of IL-1β and TNF-α cytokines	Induction the immediate-early response	Cell cultures	[132]
	MicroRNA	– Pro-inflammatory phenotype – M2 to M1 repolarization CCL2 inhibition	Lung metastasis inhibition in mouse model of breast cancer	Pre-clinical	[133,134,135]
	Exosomes	– Non-classical monocytes shift – Activation of cytotoxic NK cells	Metastasis inhibion in mice models of melanoma	Pre-clinical	[127]
	Small-molecule activators of NOD2		Metastasis inhibion in mice models of melanoma, lung, breast and colon cancers	Pre-clinical	[136]
Monocytes as therapeutical carriers	M-SMNs	Tumor cytotoxicity	Lung metastasis inhibition in mouse model of breast cancer	Pre-clinical	[137]
	CAR-M	– Tumor antigen-specific phagocytosis – Tumor clearance – Antigen presentation to T cells Cytokines and chemokines secretion	Lung metastasis inhibition in mouse model of ovarian cancer	Pre-clinical; Clinical studies Phase 1	[138,139]
Targeting Mo-MDSC	Monoclonal antibodies, peptides, small molecular inhibitors	– Blockade of Mo-MDSC functions	Reducing Mo-MDSCs	Pre-clinical; Phase II clinical trials	[140,141,142]

Note: CAR-M—chimeric antigen receptor macrophages, IL-1β—interleukin beta, M1—proinflammatory macrophages, M2—anti-inflammatory macrophages, Mo-MDSCs—monocyte myeloid-derived suppressor cells, M-SMNs—self-assembled into nanoparticles monocytes, NK—natural killer cells, TNF-α—tumor necrosis factor alpha, TEVs—tumor-derived extracellular vesicles.

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
