# Peer review of "Immune Evasion in Cancer Metastasis: An Unappreciated Role of Monocytes"

_cancers, 2025, doi:10.3390/cancers17101638_

Round 1
Reviewer 1 Report
Comments and Suggestions for Authors
This review offers a valuable and timely synthesis of current knowledge regarding the role of monocytes in cancer metastasis. The discussion of monocyte differentiation, the origin, and their dynamic interactions with tumor cells at each metastatic stage is well-articulated and supported by appropriate references.
However, I have a few suggestions that may help improve the clarity and impact of the manuscript:
-
Figure 1 Quality: The resolution of Figure 1 is insufficient and should be improved to enhance its readability. Please ensure that all elements in the figure are clearly visible when printed or viewed digitally.
-
Summary Table Suggestion: In addition to Figure 1, I recommend creating a table that summarizes the interactions between cancer cells and immune cells at each stage of metastasis. This could help readers quickly grasp the key points and compare findings across cancer types or metastatic processes.
-
Minor comment: Figure 2 might be a table.
Author Response
The authors thank the reviewer for the work done and positive feedback from the publication. All comments of reviewers were taken into account and changes were made to the text of the article, highlighted in yellow.
Comments 1 : Figure 1 Quality: The resolution of Figure 1 is insufficient and should be improved to enhance its readability. Please ensure that all elements in the figure are clearly visible when printed or viewed digitally.
Reply: In revised version of manuscript Figure 1 has been attached in high resolution (Line 148, Figure 1 revised).
Comments 2: Summary Table Suggestion: In addition to Figure 1, I recommend creating a table that summarizes the interactions between cancer cells and immune cells at each stage of metastasis. This could help readers quickly grasp the key points and compare findings across cancer types or metastatic processes.
Reply: The authors thank the reviewer for a valuable recommendation! We have added a summarizing table (Line 171-172, Table 2)
Minor comment: Figure 2 might be a table.
Reply: We have converted Figure 2 into a Table 3 (Line 381-385, Table 3).
Reviewer 2 Report
Comments and Suggestions for Authors
This review provides a timely synthesis of the current understanding of monocytes' roles in cancer metastasis. The authors present a compelling argument that monocytes—particularly their plasticity and dual function—play a more significant role in metastasis than previously appreciated. The manuscript is well-organized, with sections logically progressing from monocyte biology to clinical relevance and therapeutic strategies.The discussion of tumor-monocyte interactions, the formation of tumor hybrid cells (THCs), and the distinction between classical and non-classical monocytes is particularly strong. There are few points authors need to reconsider:
1.Although NK cell activation is mentioned in the context of non-classical monocytes, the interaction between monocytes and other immune cells (e.g., Tregs, neutrophils, dendritic cells) is not covered enough.
2. The review does not address the role of monocytes' exhaustion status in the tumor and metastatic niche. This could be relevant for linking monocyte dysfunction with immunotherapy resistance.
Author Response
The authors thank the reviewer for the work done and positive feedback from the publication. All comments of reviewers were taken into account and changes were made to the text of the article, highlighted in yellow.
Comment 1. Although NK cell activation is mentioned in the context of non-classical monocytes, the interaction between monocytes and other immune cells (e.g., Tregs, neutrophils, dendritic cells) is not covered enough.
Reply: We have added information about the interaction between monocytes with other immune cells.
Interaction with T-regs:
Line 175-176 They can attract T-regulatory cells by secreting Il-10 and TGF-b, suppress the activity of CD8 effector T-cells, and produce anti-inflammatory cytokines
Line 299 was added information, that MDSCs also produce various cytokines such as IL-6, IL-10, TGF-b, and VEGF involved in epithelial-mesenchymal transition (EMT), while MMP-2, MMP-9, and ANG2 facilitate extravasation of CTCs [95,112]. Il-10 and TGF-b promote the recruitment and activation of T-regs, which sustain immunosuppressive microenvironment.
Interaction with neutrophils:
Line 275-281: In the PMN, monocytes act together with neutrophils, also recruited by tu-mor-derived CCL2 stimuli, and facilitate tumor cell colonization. Pro-tumor neutrophils can promote tumor cell invasion through the release of proteolytic enzymes and reactive oxygen species (ROS) [101]. However, where is a contrary data about monocyte-neutrophil cooperation against tumor cells. In mice model of breast cancer, it has been demonstrated, that cooperation of CCR2 monocytes with tumoricidal TMEM173+ neutrophils target disseminated tumor cells in the lungs and prevent metastatic outgrowth [102].
Interaction with dendritic cells:
Line 101-102: In pathological conditions, circulating monocytes become the main source for replenishing the population of tissue macrophages and immature or tolerogenic dendritic cells, that lead to suppress immune response, facilitating tumor progression and metastasis [35].
Comment 2. The review does not address the role of monocytes' exhaustion status in the tumor and metastatic niche. This could be relevant for linking monocyte dysfunction with immunotherapy resistance.
Reply: Monocytes exhaustion is a phenomenon described in an in vitro experiments, for example under the influence of bacterial endotoxin lipopolysaccharide (LPS) or mycolic acid derived from Mycobacterium bovis Bacillus Calmette–Guérin (BCG) (10.3389/fimmu.2021.778830, 10.1016/j.isci.2024.108978). This dysfunctional state of monocytes plays a role in the pathogenesis of sepsis and inflammatory bowel disease (10.1016/j.celrep.2024.113894, 10.1016/j.celrep.2024.113894, 10.1038/s41598-025-86103-x, 10.3390/biomedicines12010175, 10.1016/j.humimm.2025.111257). We carefully reviewed the data on the role of exhausted monocytes in cancer development. However, there is absolutely no data on the role of exhausted monocytes in cancer pathogenesis, including metastasis. This is apparently very new data that has not yet been published and do not allow us to expand our review.
Reviewer 3 Report
Comments and Suggestions for Authors
Patysheva et al, have reviewed the role that monocytes play in the process of metastasis. Overall the review is relevant, interesting and well written and addresses an important question.
- I would like the authors to comment on their approach to searching the literature and inclusion and exclusion criteria that they used.
- It is important to note when studies are only performed in murine models; highlighting where the details are missing in patients and areas for focus in translational studies going forward, as there are some important differences between mice and humans where monocyte nomenclature/markers are concerned.
Minor
Check usage of 'recruiting' line 369; should be recruitment
Abstract line 15 - check sentence as is a little laboured- epithelial mesenchymal transition? Stem cell?
Author Response
The authors thank the reviewer for the work done and positive feedback from the publication. All comments of reviewers were taken into account and changes were made to the text of the article, highlighted in yellow.
Comment 1 : I would like the authors to comment on their approach to searching the literature and inclusion and exclusion criteria that they used.
Reply: We did not use any standard algorithms which can be described as “approach to searching”. We used the knowledge accumulated in our team in the field of monocytes and metastasis to select appropriate publish studies available in Google Scholar, PubMed and Medline databases. Here we provide several examples of key words combinations “monocytes and metastasis”, “monocytes and pre-metastatic niche”, “monocytes and metastatic cascade”, "monocytes and immune evasion". We also analyzed the literature which was cited in the publications which we identified using key words search.
Comment 2: It is important to note when studies are only performed in murine models; highlighting where the details are missing in patients and areas for focus in translational studies going forward, as there are some important differences between mice and humans where monocyte nomenclature/markers are concerned.
Reply: We have completed the section on 'Formation and Characteristics of Monocytes' by adding information about the significance of murine models for monocyte investigation and the key murine markers of monocytes (Line 71-77). Additionally, we have reviewed the information about the experimental design of the studies mentioned in the manuscript and added notes specifying that the experiments were conducted in mouse models (Line 115, 309-313, 323-326, 330-331, 333)
Minor
Check usage of 'recruiting' line 369; should be recruitment
Reply: We have changed it in revised version of manuscript
Abstract line 15 - check sentence as is a little laboured- epithelial mesenchymal transition? Stem cell?
Reply: We agree that the term “epithelial mesenchymal transition” is confusing in the abstract, we have removed it from the final version of the manuscript (Abstract, Line 15).